# Design, Synthesis, and Anticancer Effect Studies of Iridium(III) Polypyridyl Complexes against SGC-7901 Cells

**DOI:** 10.3390/molecules24173129

**Published:** 2019-08-28

**Authors:** Li-Xia Zhang, Yi-Ying Gu, Yang-Jie Wang, Lan Bai, Fan Du, Wen-Yao Zhang, Miao He, Yun-Jun Liu, Yan-Zhong Chen

**Affiliations:** 1Guangdong Provincial Key Laboratory of Advanced Drug Delivery Systems, Guangdong Pharmaceutical University, Guangzhou 510006, China; 2School of Pharmacy, Guangdong Pharmaceutical University, Guangzhou 510006, China

**Keywords:** iridium(III) complexes, antitumor activity, mitochondrial dysfunction, DNA damage, apoptosis

## Abstract

Three iridium(III) complexes ([Ir(Hppy)_2_(L)](PF_6_) (Hppy = 2-phenylpyridine, L = 5-nitrophenanthroline, NP), **1**; 5-nitro-6-amino-phenanthroline (NAP), **2**; and 5,6-diamino-phenanthroline (DAP) **3** were synthesized and characterized. The cytotoxicities of Ir(III) complexes **1**–**3** against cancer cell lines SGC-7901, A549, HeLa, Eca-109, HepG2, BEL-7402, and normal NIH 3T3 cells were investigated using the 3-(4,5-dimethylthiazol-2-yl)-2,5-diphenyl-tetrazoliumbromide (MTT) method. The results showed that the three iridium(III) complexes had moderate in vitro anti-tumor activity toward SGC-7901 cells with IC_50_ values of 3.6 ± 0.1 µM for **1**, 14.1 ± 0.5 µM for **2**, and 11.1 ± 1.3 µM for **3**. Further studies showed that **1**–**3** induce cell apoptosis/death through DNA damage, cell cycle arrest at the S or G0/G1 phase, ROS elevation, increased levels of Ca^2+^, high mitochondrial membrane depolarization, and cellular ATP depletion. Transwell and Colony-Forming assays revealed that complexes **1**–**3** can also effectively inhibit the metastasis and proliferation of tumor cells. These results demonstrate that **1**–**3** induce apoptosis in SGC-7901 cells through ROS-mediated mitochondrial damage and DNA damage pathways, as well as by inhibiting cell invasion, thereby exerting anti-tumor cell proliferation activity in vitro.

## 1. Introduction

Cancer has long been a global problem. Cancer metastasis and drug resistance are two main causes of treatment failure, with the mortality rate approaching 90% [1]. Despite great efforts in the development of treatments being made, few drugs have been approved. Among the approved drugs, small molecule–metal complexes are most desirable, due to their high availability and low cost. Cisplatin and its derivatives have been widely used in clinical practice since they were approved for the treatment of testicular and ovarian cancers [2]. However, after clinical application, they are associated with several adverse reactions, such as nausea, nephrotoxicity, bone marrow suppression, and drug resistance [3,4]. Therefore, alternatives to cisplatin are urgently needed. In recent decades, tremendous effort has been made to produce platinum-based analogues, such as oxaliplatin, for the treatment of colon and ovarian cancers [5,6]. In addition, other novel metal-based anti-tumor non-Pt complexes with better biological properties and wide anti-cancer activity ranges have been synthesized [7,8,9,10,11]. Among these, iridium(III) complexes have attracted the attention of medicinal chemists, as Ir(III) complexes have demonstrated great anti-cancer activities and shown potential as anti-tumor drugs [12,13]. 1,10-Phenanthroline has been widely used as a versatile material with many chemical uses for decades, and its structure and chemical properties have been clearly explained. In addition, 1,10-phenanthroline has been demonstrated to be a promising pharmacophore which is easily modified for the purposes of drug development [14,15]. Based on these studies and rare reports, 1,10-phenanthroline modified with an –NH_2_ or –NO_2_ group was selected as a ligand for this study of iridium metal complexes. 

Mao et al. systematically explored the iridium(III) complexes [Ir(dfppy)_2_(L_2_)](PF_6_) (dfppy = 2-(2,4-difluorophenyl) pyridine, L_2_ = 4,4′-bis(chloromethyl)-2,2′-bipyridine), which induce apoptosis in A549 cells through mitochondrial damage, cellular ATP depletion, mitochondrial respiration inhibition, and reactive oxygen species (ROS) elevation [16]. Chen and Liang et al. first demonstrated that three 5,7-dihalo-8-quinolinolato Ce(III) complexes were able to inhibit tumor growth by causing G0/G1 phase arrest [17]. Ceyda et al. investigated two Pt(Ⅱ) complexes containing saccharinate, which exhibited potent cytotoxicity toward breast (MCF-7) and colon (HCT 116) cancer cells and caused cell death/apoptosis through higher nuclear uptake, cell cycle arrest at the S phase, a significant increase in DNA double-strand breaks, apoptosis induction, elevated levels of ROS, and high mitochondrial membrane depolarization [18]. In order to investigate iridium(III) complexes with potent anti-tumor activity and to illuminate the pro-apoptotic mechanisms, we report the synthesis and characterization of three iridium(III) complexes: [Ir(Hppy)_2_(NP)](PF_6_) (Hppy = 2-phenylpyridine, NP = 5-nitro-phenanthroline, **1**) [Ir(Hppy)_2_(ANP)](PF_6_) (NAP = 5-nitro-6-amino-phenanthroline, **2**), and [Ir(Hppy)_2_(DAP)](PF_6_) (DAP = 5,6-diamino-phenanthroline, **3**) (see Scheme 1). The anti-proliferation effects of **1**–**3** on SGC-7901 cells, apoptosis, and migration are evaluated. Furthermore, subcellular localization of the complexes, activation of caspases, and mitochondrial dysfunction are investigated. Our work provides new insight for building metal anti-cancer drugs with significant anti-tumor activity.

## 2. Materials and Methods

All reagents and solvents were purchased commercially and used without further purification, unless specifically noted. Cisplatin, 1,10-phenanthroline, hydrazine hydrate, palladium carbon reagent, and hydroxylamine hydrochloride were purchased from the Guangzhou Chemical Reagent Factory. IrCl_3_·3H_2_O was obtained from the Kunming Institution of Precious Metals. Propidium iodide (PI), MTT, MitoTracker® Deep Red FM (150 nM, λ_ex_ = 575 nm, λ_em_ = 600 nm), 2′,7′-dichlorodihydro-fluorescein diacetate (DCHF-DA, 10 µM, λ_ex_ = 488 nm, λ_em_ = 525 nm), carbonyl cyanide m-chlorophenylhydrazine (CCCP, 10 mM), Fluo-3AM (Molecular Probes, Eugene, OR, 2.5 µM, λ_ex_ = 488 nm, λ_em_ = 525 nm), N-acetylcysteine (NAC), 5,5′,6,6′-tetrachloro-1,1′,3,3′-tetraethylbenzimidazolylcarbocyanine iodide (JC-1), and ethidium bromide (EB) were obtained from the Beyotime Institute of Biotechnology (Shanghai, China). Stock solutions of the complexes (20 mM) were prepared in DMSO and stored at 4 °C (final concentration of DMSO less than 0.1% *v*/*v*). Deionized water was used in all experiments. 

### 2.1. Physical Measurements

Microanalyses (C, H, and N) were performed using a PerkinElmer 240Q elemental analyzer. Electrospray ionization mass spectroscopic (ESI–MS) analyses were performed using an LCQ system (Finnigan MAT, Waltham, MA, USA), where the mass spectra were recorded in positive mode and acetonitrile was used as the mobile phase. The spray voltage, tube lens offset, capillary voltage, and capillary temperature were set at 4.50 kV, 30.00 V, 23.00 V, and 200 °C, respectively, and the quoted *m/z* values were for the major peaks in the isotope distribution. ^1^H NMR and ^13^C NMR spectra were acquired at room temperature on a Varian-500 spectrometer (500 MHz) with DMSO-d_6_ as the solvent and tetramethylsilane (TMS) as an internal standard. UV-visible spectra were obtained on a Shimadzu UV-3101PC spectrophotometer. Fluorescence measurements were carried out on a Shimadzu RF-4500 fluorescence/phosphorescence spectrophotometer at room temperature.

### 2.2. Synthesis of Iridium(III) Complexes

#### 2.2.1. Synthesis of [Ir(Hppy)_2_(NP)]PF_6_ (**1**)

Complex **1** was prepared by a conventional synthetic method, in which a mixture of dichloromethane and methanol (42 mL, 2:1) was added to a flask containing [Ir(Hppy)_2_Cl]_2_ (0.323 g, 0.30 mmol) and NP (0.135 g, 0.60 mmol) [19]. The mixture was refluxed for 6 h under argon to give a red brown solution. After cooling, a bright red precipitate was obtained by dropwise addition of concentrated NH_4_PF_6_ aqueous solution with stirring at room temperature for 2 h. The crude product was purified by column chromatography on alumina eluted with dichloromethane–acetone (1:3, *v*/*v*). The red band was collected, the solvent was evaporated under the reduced pressure, and a brown-yellow powder was obtained. Yield: 86%. Anal. Calc for C_34_H_23_F_6_N_5_O_2_PIr: C, 46.90; H, 2.66; N, 8.04%. Found: C, 46.81; H, 2.72; N, 8.12%. ^1^H NMR (500 MHz, DMSO-*d*_6_): 9.46 (s, 1H), 9.20 (d, 1H, *J* = 8.0 Hz), 9.12 (d, 1H, *J* = 7.5 Hz), 8.34 (dd, 2H, *J* = 5.5, *J* = 6.0 Hz), 8.26 (d, 2H, *J* = 8.0 Hz), 8.19–8.15 (m, 2H), 7.95 (d, 2H, *J* = 8.0 Hz), 7.88 (t, 2H, *J* = 7.5 Hz), 7.52 (dd, 2H, *J* = 6.0, *J* = 6.0 Hz), 7.06 (t, 2H, *J* = 7.5 Hz), 7.01–6.94 (m, 5H), 6.26 (d, 2H, *J* = 7.5 Hz). ^13^C NMR (125 Hz, DMSO-*d*_6_): 166.70, 153.41, 151.88, 149.56, 149.11, 147.87, 146.79, 144.96, 144.04, 140.80, 138.87, 135.24, 131.23, 130.33, 128.52, 128.34, 127.45, 125.14, 123.97, 123.89, 122.60, 120.03. ESI-MS (CH_3_CN): *m/z* 725.9 ([M − PF_6_]^+^).

#### 2.2.2. Synthesis of [Ir(Hppy)_2_(NAP)]PF_6_ (**2**)

**2** was synthesized through an identical method to **1**, with NAP (0.144 g, 0.60 mmol) [20] in place of NP. Yield: 81%. Anal. Calc for C_34_H_24_F_6_N_6_O_2_PIr: C, 46.10; H, 2.73; N, 9.49%. Found: C, 46.23; H, 2.82; N, 9.36%. ^1^H NMR: (500 MHz, DMSO-*d_6_*): 9.38 (d, 1H, *J* = 8.0 Hz), 8.93 (dd, 3H, *J* = 8.0 Hz), 8.26 (d, 3H, *J* = 8.0 Hz), 8.06 (q, 1H, *J* = 4.0 Hz,), 7.95–7.86 (m, 6H), 7.54 (t, 2H, *J* = 8.0 Hz), 7.08–7.01 (m, 4H), 6.94 (d, 2H, *J* = 8.0 Hz,), 6.24 (t, 2H, *J* = 8.0 Hz). ^13^C NMR (125 MHz, DMSO-*d*_6_): 166.76, 152.81, 150.03, 149.49, 149.33, 148.81, 146.56, 144.39, 144.00, 143.94, 140.39, 138.80, 135.57, 132.58, 131.23, 131.11, 130.31, 130.28, 128.07, 127.52, 127.22, 125.13, 125.09, 124.86, 123.95, 123.84, 122.49, 122.44, 121.80, 120.02, 119.97. ESI-MS (CH_3_CN): *m/z* 741.0 ([M − PF_6_]^+^).

#### 2.2.3. Synthesis of [Ir(Hppy)_2_(DAP)]PF_6_ (**3**)

**3** was synthesized in accordance with a method described in the literature [21]. Yield: 80%. Anal. Calc for C_34_H_26_F_6_N_6_PIr: C, 47.72; H, 3.06; N, 9.82%. Found: C, 47.78; H, 3.11; N, 9.73%. ESI-MS (CH_3_CN): *m/z* 711.5 ([M − PF_6_]^+^).

### 2.3. Cell Culture

The lung carcinoma cell line A549, the cervical cancer cell line HeLa, the esophageal cancer cell line Eca-109, and the human hepatocellular carcinoma cell line BEL-7402 were purchased from the cell bank of the Cell Institute of Sinica Academia Shanghai (Shanghai, China). The gastric adenocarcinoma cell line SGC-7901, the hepatocellular carcinoma cell line HepG2, and the mouse embryonic fibroblast cell line NIH 3T3 were obtained from the Experimental Animal Center of Sun Yat-Sen University (Guangzhou, China). The BEL-7402 and SGC-7901 cell lines were cultured in Roswell Park Memorial Institute Medium (RPMI-1640); and A549, Eca-109, HepG2, and NIH 3T3 cells were grown in Dulbecco’s Modified Eagle’s Medium (DMEM), including 10% fetal bovine serum (FBS), 100 units mL^−1^ penicillin/streptomycin mixture, and 2.0 g/L of NaHCO_3_. Cell passage experiments were performed with trypsin every 2 days to maintain exponential growth. All cells were cultured until they reached logarithmic growth phase, unless specifically noted.

### 2.4. Oil–Water Partition Coefficient Determination

The pH of the cells cultured in vitro was approximately 7.4. With this in mind, we evaluated the lipophilicity of the iridium complexes by detecting the partition coefficient at pH = 7.4, using the flask-shake method. Each drug was dissolved in pre-saturated octanol at final concentrations of 0.02 and 0.03 mM. Aliquots of 5 mL of drug solution and distilled water (previously saturated with octanol) were transferred to a 25 mL flask and shaked at 25 °C for 24 h. After the shaking was completed, the solution was allowed to stand for 48 h to fully equilibrate the aqueous phase and the organic phase. The concentration of the complexes in the aqueous phase was then measured by UV-vis spectroscopy at the maximum ultraviolet absorption (262 nm for **1**, 249 nm for **2**, and 251 nm for **3**). The evaluation of each sample was repeated three times.

### 2.5. Cell Uptake

SGC-7901 cells were plated in a 12-well plate (Costar, Corning Corp, New York, NY, USA) at a density of 1.5 × 10^5^ cells per well. After overnight of cultivation, the medium was removed and replaced by the different concentrations of the complexes for 3 h at 37 °C Then, the plate was washed three times with PBS buffers to remove the residual complexes and the cells were further incubated with DAPI for 20 min. After that, the cells were washed twice with ice-cold PBS to remove residual DAPI and observed under fluorescence microscope.

### 2.6. In Vitro Cytotoxicity Assay 

Cell cytotoxicity following treatment with the iridium(III) complexes was detected by MTT assay [22,23]. The cells were seeded into 96-well micro-assay culture plates (1 × 10^4^ cells per well) and grown overnight in a 5% CO_2_ incubator at 37 °C. Then, the cell lines were treated with gradient concentrations of the complexes **1**–**3**. After treatment for 48 h, MTT solution (5 mg mL^−1^, 20 mL/well) dissolved in phosphate buffer was added to each well and the cells were further cultured for 4 h. Upon completion of incubation, the solvent was removed and 100 µL/well of buffer containing dimethylformamide (50%) and sodium dodecyl sulfate (20%) was added to dissolve the produced formazan. The absorbance of each well was measured with a microplate reader at a wavelength of 490 nm. The IC_50_ values were determined from the plots of viability versus concentration used to treat the cells. Three independent experiments were performed to obtain the mean values.

### 2.7. Colony-Forming Assay

The SGC-7901 cells were seeded at a cell density of 500 cells per well in a 6-well plate. After 24 h, an equal amount of solvent or iridium complexes was added into the medium for another 24 h. Then, the medium containing the tested complexes was removed and replaced with fresh medium. The number of the colonies was counted on the 8th day after seeding and then fixed with paraformaldehyde for 20 min. The colonies were photographed under an inverted microscope after staining with 0.1% (*w*/*v*) crystal violet for 20 min. Each experiment was carried out in triplicate.

### 2.8. Analysis of Cell Invasion

To analyze the efficacy of the iridium(III) complexes in preventing migration and offsetting the invasion potential of SGC-7901 cells, 24-well Transwell chambers coated with matrigel were used. The cells were trypsinized and re-suspended in serum-free RPMI-1640 at a density of 1.8 × 10^5^ cells/mL. Then, 100 μL of the cell suspension and different concentrations of the complexes were placed in the top chambers of the Transwell plates, and 500 μL of RPMI-1640 containing 20% FBS was added to the lower chambers. After co-incubation for 24 h, non-invading cells in the top chambers of the Transwell plates were scraped away by cotton swabs, and invaded cells were fixed with paraformaldehyde for 15 min and stained with crystal violet solution (0.1%) for 20 min. The cells that invaded the underside of the Transwell plates were photographed with a microscope.

### 2.9. Comet Assay

Cells in the logarithmic growth phase were treated with the iridium(III) complexes for 24 h. After treatment, cells were trypsinized and re-suspended in low-melting agarose (10 mg/mL) dissolved in PBS. Normal melting point agarose (0.5 mg/mL) was prepared and pre-coated on a glass slide, and then the mixture of cells and agarose was spread onto the slide. The slides were covered with low-melting agarose (10 mg/mL) and solidified at 4 °C for 10 min, before being kept overnight in cell lysate solution (2.5 M NaCl, 0.1 M EDTA, 10 mM Tris base, 10% DMSO, 1% Triton X-100, pH 10) at 4 °C The slides were set in an electrophoresis tank containing freshly prepared alkaline buffer (300 mM NaOH and 1 mM EDTA, 4 °C, pH > 13) and immersed for 20 min for DNA unwinding. Electrophoresis was performed at 25 V (300 mA) for 20 min in the above buffer. After electrophoresis, the slides were immersed in the neutralization buffer (400 mM Tris, HCl, pH 7.5) for 10 min and then dehydrated with 70% ethanol. Finally, the slides were stained with EB (20 mg/mL in distilled water) and subsequently observed under a fluorescence microscope.

### 2.10. Apoptosis Analyses

A total of 1 mL of SGC-7901 cells at 2 × 10^6^ cells/mL was seeded in a six-well plate and incubated overnight in a CO_2_ incubator at 37 °C. Then, the cells were treated with **1**, **2**, or **3** for 24 h. After treatment, the cells were harvested, washed three times with ice-cold PBS, and then re-suspended in 195 μL of 1 × Annexin V-FITC binding buffer. Next, 10 μL of PI and 5 μL of Annexin-FITC were sequentially added to each sample, stained in the dark for 20 min, and analyzed by a FACS Calibur flow cytometer (Beckman Dickinson & Co., Franklin Lakes, NJ, USA). Annexin-V positive cells were considered apoptotic.

### 2.11. Measurement of Reactive Oxygen Species

2,7-Dichloro-dihydrofluorescein diacetate (DCFH-DA) was used as a fluorescence probe to investigate the changes in intracellular ROS levels. SGC-7901 cells were placed in a 12-well plate at a density of 1.5 × 10^5^ per well. After 24 h, the medium in the wells was replaced with a medium containing the corresponding concentration of **1**–**3**, and the cells were incubated for 24 h. Finally, the cell pellets were suspended in PBS and imaged under the ImageXpress Micro XLS system (MD company, US). The DCF fluorescence intensity was calculated by flow cytometry.

### 2.12. Determination of Intracellular Ca^2+^ Levels

SGC-7901 cells in the logarithmic growth phase were seeded in 12-well microassay culture plates (1.5 × 10^5^ cells per well), incubated overnight, and then treated with different concentrations of the tested complexes (4.0 µM **1**, 15.0 μM **2**, and 15.0 μM **3**) for 24 h. The control wells were prepared by the addition of solvent (10 μL PBS buffer). The concentration of Ca^2+^ was detected using the Ca^2+^-sensitive fluorescent probe Fluo-3AM. Upon completion of incubation, the medium was removed and the wells were washed twice with PBS buffer, following which the cells were dyed with Fluo-3AM for 30 min. After dyeing, the cells were washed again and stained with Hoechst 33258 (10 μg mL^−1^) for 30 min. The remaining dye was removed and the cells were observed under an ImageXpress Micro XLS system.

### 2.13. Localization at the Mitochondria of the Complexes

The SGC-7901 cells were plated in a 12-well plate (Costar, Corning Corp, New York) at a density of 1.5 × 10^5^ cells per well. After incubation for one night, the cells were washed twice with PBS buffer and then treated with different concentrations of the complexes for 8 h at 37 °C. After treatment, the cells were washed twice with PBS and were further incubated with Mito-tracker Red for 30 min. The cells were then carefully washed twice with ice-cold PBS to remove residual stain and submitted to the ImageXpress Micro XLS system for inspection.

### 2.14. Mitochondrial Membrane Potential Assay (∆Ψm)

The SGC-7901 cells were seeded into a 12-well plate at a density of 1.5 × 10^5^ cells per well and incubated to attach at 37 °C in 5% CO_2_. Overnight, the cells were treated with complexes **1**–**3** for 24 h. CCCP was used as a positive control. Then, the cells were incubated with JC-1 for 30 min at room temperature, after which the remaining dye was washed away with PBS. The samples were then observed under the ImageXpress Micro XLS system and flow cytometry was employed to quantify the fluorescence intensity.

### 2.15. ATP Quantification Assay

Approximately 2.4 × 10^6^ cells were taken, evenly planted in a 6-well plate, and incubated overnight in an incubator at 37 °C. On the next day, the cells were exposed to different concentrations of compounds for 24 h in triplicate. Following exposure, 200 μL of cell lysis buffer was added to each well of the cell plate and lysed on ice. To measure ATP, 20 μL of the lysed cell suspension was added to 100 μL of a luciferin–luciferase reagent. Light emission was measured by a luminometer. ATP levels were determined using a standard curve. 

### 2.16. Cell Cycle Assay

After treatment with 4.0 μM **1**, 15.0 μM **2****,** and 15.0 μM **3** for 48 h, the SGC-7901 cells were trypsinized and centrifuged (10 min at 800× *g*) and fixed with 75% cold ethanol overnight at 4 °C. Then, each sample was washed twice with cold PBS, re-suspended in 190 μL of PBS buffer containing 4 μL of PI (1 mg/mL), 4 μL of RnaseA (10 mg/mL) and 0.2 μL of Tritonx-100, and stained in the dark for 30 min. Finally, the cells were analyzed by flow cytometry

### 2.17. Western Blot Analysis

The SGC-7901 cancer cells were seeded on 6-well plates at a density of 5 × 10^5^ cells per well. Following incubation for 24 h, the cells were treated with **1**–**3** for 24 h. Next, total proteins were extracted from the cells after incubation in lysis radioimmunoprecipitation assay (RIPA) buffer for 30 min on ice. After centrifugation (13,000× *g*, 4 °C, 20 min) of the cell suspension, the supernatants were obtained. Protein concentrations were determined using a BCA (bicinchoninic acid) protein assay kit. Equal amounts of denatured proteins were fractionated by electrophoresis on SDS-polyacrylamide gel at 50 V for 3 h. A pre-stained color protein marker was used as the internal reference. The gels were then segmented, transferred to polyvinylidene fluoride (PVDF) membranes (Millipore, Burlington, MA, USA) and then blocked with 5% non-fat milk in TBST buffer (150 mM NaCl, 20 mM Tris-HCl, 0.05% Tween 20, pH = 8.0) for 2 h. After blocking, the membranes were hybridized overnight at 4 °C with primary antibodies (diluted 1:5000) against PARP, Cleaved PARP, Caspase 3, Bax, Bak, Bcl-2, and β-actin. Further hybridization with the secondary antibodies (diluted 1:5000) conjugated with horseradish peroxidase was performed after washing the membranes three times in TBST buffer with shaking. Each protein was visualized using an enhanced chemiluminescence (ECL) detection kit, in accordance with the manufacturer’s instructions. Finally, the blot images were photographed by a computerized imaging system.

## 3. Results and Discussion

### 3.1. Synthesis and Characterization

The structures of **1**, **2**, and **3** are illustrated in Scheme 1. The ligands NP, NAP, and DAP were prepared, in accordance with methods described in the literature [19,20]. Ir(III) complexes were obtained in relatively high yields by refluxing [Ir(Hppy)_2_Cl]_2_ and the corresponding ligands at a molar ratio of 1:2. The expected Ir(III) complexes were purified by column chromatography on neutral alumina and characterized by elemental analysis, ^1^H NMR, ^13^C NMR, and ESI-MS. In the ESI-MS spectra, the desired signals of [M − PF_6_]^+^ were discovered at *m/z* values of 725.9 for **1** and 741.0 for **2** and 711.5 for **3**. In the spectra of ^1^H NMR, the shifts of 0.17 for H_a_ (NP, 9.37 [19]; **1**, 9.20 ppm) and 0.19 for H_c_ (NP, 9.31 [19]; **1**, 9.12 ppm) were observed. A great change of 0.76 for H_d_ was also found. For NAP and complex **2**, the changes of 0.08 ppm for H_a_ (NAP, 9.30 [20]; **2**, 9.38 ppm) and 0.18 ppm for H_c_ (NAP, 9.20 [20]; **2**, 9.38 ppm) were discovered. These changes in the chemical shift of proton indicate that the complexes were successfully synthesized. The absorption and emission spectra of **1**, **2**, and **3** were measured at room temperature. The UV-Vis absorption spectra of the complexes in PBS buffer are presented in Figure 1A; a strong high-energy absorption band was shown in the 240–300 nm region and a maximum absorption peak was exhibited at wavelengths of 262, 249, and 251 nm for **1**, **2**, and **3,** respectively. These peaks were assigned to the intraligand π-π* transitions. In addition, the luminescence spectra of **1**, **2**, and **3** in PBS buffer under excitation at a specific wavelength are shown in Figure 1B, with the maximum emission wavelength at 604 nm for **1**, 611 nm for **2**, and 605 nm for **3**. In the assay of bioactivity, the emission wavelengths of 525 nm for ROS (2′,7′-dichlorodihydro-fluorescein diacetate (DCHF-DA, 10 μM, *λ*_ex_= 488 nm), 529 nm (JC-1, 5′,6,6′-tetrachloro-1,1′,3,3′-tetraethylimidacarbocyanine iodide, *λ*_ex_ = 514 nm) for mitochondrial membrane potential and 525 nm (Fluo-3AM, *λ*_ex_ = 488 nm) for Ca^2+^ are less than 580 nm. Therefore, the green fluorescence of the complexes has no disturbance on the fluorescence of dyes.

### 3.2. Determination of Lipophilicity

The lipophilicity (log*P_o/w_* values) of a compound has a strong influence on its cellular uptake and localization and is a major physicochemical property in drug discovery and development [24,25]. The lipophilicity of complexes **1**–**3** was determined by the flask-shaking method, and the log*P_o/w_* values are depicted in Table 1. The log*P_o/w_* values followed the order of **2** (1.33) > **1** (1.01) > **3** (0.78). **1** (1.01) and **2** (1.33) were found to be much more lipophilic than their non-nitro counterpart [Ir(Hppy)_2_(DAP)](PF_6_) (0.78), which reflects the highly hydrophobic capacity of the nitro group.

### 3.3. Cellular Uptake

After uptake by the cells, the complexes could accumulate in different subcellular structures of the cells. To detect the cell international process and the ultimate intracellular localization, DAPI, a nuclear-specific dye, was used to explore the subcellular structure in which the complexes **1**, **2** and **3** locate. After the treatment with the complexes for 3 h, SGC-7901 cells were observed under a fluorescence microscope, and the results were shown in Figure 2. The complexes appeared as green fluorescence, whereas nucleus stained with DAPI exhibited blue fluorescence. As Figure 2 depicted, the blue fluorescence of DAPI and the green fluorescence of the complexes were overlapped, indicating that the complexes were mainly accumulated in the cytoplasm at 3 h.

### 3.4. Cytotoxic Activity In Vitro

To examine the activity of **1**, **2**, and **3** as potential anti-cancer drugs, their cytotoxicity toward a series of human cancer cell lines and a normal cell line was evaluated at gradient concentrations using the MTT method. The IC_50_ values (the concentration at which 50% of cell growth is inhibited) for **1**, **2**, and **3** against these cell lines are summarized in Table 1. It is notable that, in comparison with other cancer cells, the Ir(III) complexes exhibited sensitive activity to SGC-7901 cells, demonstrating potent cytotoxicity. The IC_50_ value of **1** (3.6 μM) against SGC-7901 cells was lower than those of **2** and **3**, indicating that **1** has better cytotoxicity than **2** and **3**. Although **2** exhibited the largest log*P_o/w_* value, **1** showed the highest cytotoxicity toward the selected cell lines. Therefore, the lipophilicity of the complexes is not consistent with the cytotoxicity in vitro. Based on these phenomena, we infer that there are some correlations between the special molecular structures of the complexes and their anti-tumor activities in vitro. Complexes **2** and **3**, containing –NH_2_ and –NO_2_ groups, exhibited low cytotoxicity compared with **1**, which only contains –NO_2_; therefore, we conjecture that the –NH_2_ group may prevent **2** and **3** from entering the cell membrane, therefore resulting in weaker activity. In addition, we also determined the cytotoxic activity of free ligands NP, NAP, DAP and starting material [Ir(Hppy)_2_Cl]_2_ toward the above cancer cells. As expectation, free ligands show moderate cytotoxic activity. Their cytotoxicities are lower than their complexes against SGC-7901 cells. However, in the inhibition of cell growth of A549, HeLa and HepG2 cells, the ligands show higher effect than their corresponding complexes under the same condition. This may be attributed to the different sensitivity of ligands and their complexes toward different cancer cells. It is unexpected to find that the starting material [Ir(Hppy)_2_Cl]_2_ displays higher cytotoxicity than all the ligands and their complexes against all the selected cancer cell lines. We infer that [Ir(Hppy)_2_Cl]_2_ exhibits different anticancer mechanism compared with the complexes. Because complexes **1**, **2,** and **3** had high sensitivity to SGC-7901 cells, this cell line was selected as the research subject for anti-tumor activity. In the biological activity assays, we used near IC_50_ values as concentration of the complexes, namely, 4.0 µM for **1**, 15.0 µM for **2** and **3**, respectively.

### 3.5. Inhibition of Colony Formation

Colony formation is one of the key features of malignant cancer cells. Metastatic cells must continue to multiply at new sites to cause tumor metastasis [12]. Thus, we evaluated the inhibition of colony formation induced by 4.0 µM **1**, 15.0 µM **2**, and 15.0 µM **3**. As shown in Figure 3, after SGC-7901 cells (a) were incubated with **1** (b), 2 (c) and **3** (d) for 8 days, significant inhibition of colony formation was clearly observed, which was manifested as a sharp decrease in cell density. This demonstrates that **1**–**3** exerted an anti-tumor effect by inhibiting proliferation in vitro.

### 3.6. Transwell Migration Assay

Part of the reason why malignant tumors are difficult to cure is their strong ability to invade and metastasize [26]. In order to evaluate the in vitro anti-invasive capacities of **1**–**3**, Transwell assays were performed. The migrated SGC-7901 cells were stained with crystal violet and recorded by inverted microscope after treatment of SGC-7901 cells (a) with **1** (b, 4.0 µM), **2** (c, 15.0 µM), and **3** (d, 15.0 µM) for 24 h. As is apparent from Figure 4A, the iridium complexes dramatically decreased cell migration compared with the control group, in which 100% of cells were considered to have migrated. As shown in Figure 4B, the inhibitory percentages of **1**, **2**, and **3** were found to be 51.09%, 46.01%, and 48.90%, respectively. These results reveal that the iridium(III) complexes inhibited the metastasis of cancer cells while exerting anti-tumor activity.

### 3.7. Comet Assay

DNA is a key biological target for anti-tumor drugs. The design and synthesis of DNA-targeted metal-based anti-cancer agents with potential anti-cancer activities have recently received great attention [27]. This section describes the use of a comet assay to detect DNA damage. After exposure to different complexes for 24 h, DNA damage in SGC-7901 cells was clearly observed using fluorescence microscopy (×20 magnification). The length of the comet tail was utilized as a parameter for DNA damage [28]. As can be seen in Figure 5, the vehicle control-treated cells showed no or negligible DNA damage. In contrast, the migration distance of DNA was dramatically increased after the incubation of SGC-7901 cells with any of the complexes. The results indicate that the three iridium complexes could cause intracellular DNA fragmentation; similar phenomena have been commonly observed in other metal-based complexes [29,30]. The results show that the complexes can cause DNA damage, which is a hallmark of apoptosis.

### 3.8. Apoptosis Studies

Apoptosis is a common cell death pathway in metal-based anti-cancer complex therapy [31]. To further evaluate whether the studied complexes can induce apoptosis, SGC-7901 cells were treated with them for 48 h. Eversion of the phosphatidylserine membrane is a feature of cells undergoing apoptosis. Therefore, the Annexin-V/PI double staining test was performed and then submitted to a flow cytometer for detection [32]. In contrast to vehicle control-treated cells (Figure 6a), which showed an early apoptotic (EA) rate of 2.84%, the percentages of SGC-7901 cells incubated with 4.0 µM **1** (b), 15.0 µM **2** (c), or **3** (d) were 29.9%, 14.8%, and 13.7%, respectively. Whereas the percentages including early and late apoptotic follow the order of **3** (41.8%) > **1** (37.07%) > **2** (35.6%). The results reveal that these complexes can indeed induce apoptosis, with **1** displaying the highest apoptotic effect.

### 3.9. Iridium(III) Complexes Prompt ROS Production

A change in ∆Ψm is usually accompanied by an increase in the ROS level. Thus, in order to detect the intracellular ROS production, the control and treated SGC-7901 cells were stained with a DCFH-DA fluorescent probe and submitted to the ImageXpress Micro XLS system. As can be seen in Figure 7A, compared with the control group (a), after treatment with Rosup (b, positive control) and **1** (c), **2** (d), and **3** (e) for 24 h, there was increased green fluorescence intensity, indicating an increase in the intracellular ROS level.

NAC inhibited the production of reactive oxygen species, thereby restricting changes in the mitochondrial membrane potential. To further verify whether increased ROS accumulation was related to the anti-tumor activity in vitro, a quantitative analysis of DCF fluorescence intensity was performed, according to the flow chart (Figure 7B), there was a statistically obvious increase after treatment with 4.0 µM **1** (c), 15.0 µM **2** (e), and 15.0 µM **3** (g) compared to the control (a). However, when the cells (b) were treated with the combinations 4.0 µM **1** + NAC (d), 15.0 µM **2** + NAC (f), and 15.0 µM **3** + NAC (h), the ROS yield was dramatically reduced and the fluorescence intensity was obviously decreased compared with the values of the complexes alone. Owing to the complexes emitting green fluorescence, to detect whether the green fluorescence has influence on ROS levels, we also determined the fluorescence intensity in the presence of cell and complexes without dyes. The weak fluorescence intensity is 780 for **1** (4.0 M) + cell, 507 for **2** (15.0 M) + cell and 1120 for **3** (15.0 M) + cell, respectively. Hence, the green fluorescence emitted by the complexes have no obvious effect on the intracellular ROS levels. 

### 3.10. Detection of Intracellular Ca^2+^ Levels

To further explore whether the accumulation of intracellular ROS is accompanied by an increase in intracellular Ca^2+^ levels, the Ca^2+^ concentration was measured using the ImageXpress Micro XLS system with the Fluo-3AM fluorescent probe. As can be seen in Figure 8, after treating the cells with 4.0 µM **1**, 15.0 µM **2**, or 15.0 µM **3** for 24 h, brilliant green fluorescence, which overlapped with DAPI fluorescence over a large area, was observed in the cells. Contrary to these results, the untreated cells (controls) revealed weak green fluorescence under the same conditions. Mitochondrial Ca^2+^ overload is a key signal of mitochondrial dysfunction. According to the literature [33,34], intracellular Ca^2+^ overload triggered by ox-LDL will induce calpain-mediated mitochondrial permeability transition pore (mPTP) opening and cell apoptosis. It is clear that complex-induced mitochondrial damage occurs through a Ca^2+^-dependent pathway. In addition to this, Michelle N. Sullivan et al. reported that ROS produced by lipid peroxidation metabolites promote Ca^2+^ influx by activating the TRP1 channel, thereby improving the expansion of cerebral arteries [35]. Yi et al. also found a significant decrease in ROS-dependent Ca^2+^ levels in the presence of NAC, which was manifested as a decline in fluorescence intensity [36]. It is worth noting that ROS accumulation precedes mitochondrial membrane changes and other typical apoptotic events [37]. In combination with our previous experiments, ROS may act as an upstream regulator of complex-induced Ca^2+^ influx. 

### 3.11. Cellular Localization and Mitochondrial Damage Analysis

The three Ir(III) complexes can emit green fluorescence, and we utilized this fluorescence property to assay co-localization, using Mito Tracker Deep Red (MTDR) as a red fluorescence probe for the mitochondria. The fluorescence microscopic images show that the complexes effectively infiltrated into SGC-7901 cells after 8 h of incubation (as indicated by the bright green fluorescence in the cytosol) (Figure 9A). Furthermore, under the same conditions, the images of Ir(III) complexes almost completely overlap with those of Mito Tracker Deep Red, indicating that the Ir(III) complexes can localize in the mitochondria. Thus, the localization experiment demonstrated the feasibility of the Ir(III) complexes to target the mitochondria and indicated that the anti-tumor properties may be derived from mitochondria-mediated cell death.

According to the literature [38,39] and our previous work [40,41], we think that the complexes enter into the mitochondrial, then the complexes cause an increase of intracellular ROS levels, which caused the depolarization of mitochondrial membrane. The overproduction of ROS triggers the change in the mitochondrial membrane potential (ΔΨm), which might have instigated the intrinsic pathway of apoptosis. Alterations of ΔΨm have been commonly used to assess mitochondrial dysfunction [42]. To investigate whether the selective localization of **1**–**3** into the mitochondria could lead to mitochondrial dysfunction, the fluorescent probe JC-1 was used to determine the changes in ΔΨm by using the ImageXpress Micro XLS system and flow cytometry. As shown in Figure 9B, the decrease in ΔΨm can be easily observed by the transition from red to green fluorescence of JC-1, indicating the disruption of mitochondrial function. Once the integrity of the mitochondrial membrane is disrupted, pro-apoptotic factors are released from the mitochondria into the cytoplasm, initiating a cascade of apoptotic pathways [43]. These results demonstrate the destruction of normal mitochondrial function after co-hatch of SGC-7901 cells with the different complexes, which may have triggered a series of apoptotic signals. 

To further explore the correlation between ROS production and mitochondrial membrane potential, the changes in ΔΨm were also detected through an evaluation of the ratio of red/green fluorescence intensity by flow cytometry. As depicted in Figure 9C, when the SGC-7901 cells (a) were incubated with carbonyl cyanide m-chlorophenylhydrazine (CCCP, c, positive control), 4.0 µM of **1** (d), 15.0 µM of **2** (f), or 15.0 µM of **3** (h) for 24 h, the ratios of red versus green fluorescence were 12.03, 3.29, 0.43, 7.12, and 1.49, respectively. However, when SGC-7901 cells (b) were incubated with 4.0 µM of **1** (e), 15.0 µM of **2** (g), or 15.0 µM of **3** (i) in the presence of N-acetylcysteine (NAC, an antioxidant with cytoprotective effects against ROS-induced cell death) for 24 h, the ratios of red versus green fluorescence were 19.40, 1.86, 17.99, and 14.09, respectively, indicating a greater decrease in ROS levels with NAC than with the complexes alone. The results showed that NAC acted as a reactive oxygen species inhibitor, which dramatically prevented a decrease in the mitochondrial membrane potential. In other words, ROS play a vital role in the process of mitochondrial membrane potential changes. Therefore, we infer that the mitochondrial damage induced by **1**–**3** may have been caused by the accumulation of intracellular ROS.

### 3.12. Determination of ATP

Adenosine triphosphate (ATP), an energy-rich phosphate compound that is present in cells, plays a vital role in intracellular metabolism by directly providing energy for most cellular events [44]. ATP is a multi-functional nucleotide involved in a series of biological processes, and abnormal levels of ATP are closely related to various diseases such as malignant tumors; thus, ATP has been regarded as a critical indicator for cell viability and cell injury [45,46]. Loss of mitochondrial membrane potential, which prevents mitochondrial ATP production, has been shown to be a signal of cell apoptosis [47]. Therefore, intracellular ATP concentrations were assessed as an apoptosis-related metabolic parameter in the SGC-7901 cells. In the control cells, the ATP amount was 951.46 nM. After exposure of SGC-7901 cells to 4.0 µM of **1**, 15.0 µM of **2,** or 15.0 µM of **3** for 24 h, the ATP amounts were 913.91, 880.48, and 945.02 nM, respectively. Compared with the control group, intracellular ATP levels were significantly reduced after incubation of the cells with the iridium(III) complexes for 24 h. Mitochondrial damage was also confirmed by a decrease in the ATP content of complex-treated cells.

### 3.13. Cell Cycle Distribution

The cell cycle is closely related to the proliferation and development of tumor cells. According to the literature, cell cycle arrest is commonly observed in many metal-based complex anti-tumor drugs [48,49]. Meanwhile, the anti-cancer activity of many anti-cancer complexes has been associated with cell cycle disturbances [50]. Anti-tumor drugs induce cell cycle arrest by inhibiting DNA synthesis and enhancing DNA damage, ultimately leading tumor cell to apoptosis [51,52]. To investigate whether the anti-tumor activities of **1**–**3** are due to DNA damage-induced cell cycle arrest, SGC-7901 cells were stained with the fluorescent probe PI and, subsequently, detected by flow cytometry. As shown in Figure 10, the percentage of cells at the S phase was 37.46% in the control group. After treatment with **1** (4.0 μM) or **3** (15.0 μM) for 24 h, the S phase percentage increased to 43.02% and 44.71%, respectively. At the same time, the percentage of G2/M phase cells treated with **1** or **3** decreased to 6.70% and 10.50%, respectively, compared to 14.92% in the control. Therefore, **1** and **3** inhibited cell proliferation at the S phase. However, in SGC-7901 cells which were exposed to **2** (15.0 μM) for 24 h, an increase of 9.03% was observed in the cells at the G0/G1 phase, indicating the induction of cell cycle arrest at G0/G1 by **2**. These results reveal that **1**, **2** and **3** can inhibit cell proliferation at the S or G0/G1 phases.

### 3.14. Western Blot Analysis

During apoptosis, mitochondrial membrane permeability (MMP) is increased and proapoptotic factors are released into the cytoplasm. Bcl-2 and Bax are two well-known members of the Bcl-2 protein family that regulate the balance between proliferation and apoptosis in apoptotic cells [53]. A variety of cytotoxic stimuli are involved in the mitochondrial pathway regulated by the Bcl-2 protein family. These stimuli disrupt the integrity of the mitochondrial membrane by activating promoter downregulating pro-survival Bcl-2-like proteins and upregulating the pro-apoptotic proteins Bax and Bak [54]. Apoptotic factors are released from the damaged mitochondria to the cytoplasm, thereby activating the downstream protein caspase-3 of the caspase family, and apoptosis progresses into an irreversible phase [55]. To further explore the molecular mechanisms of apoptosis induced by the complexes, Western blotting was used to detect the levels of protein expression. As shown in Figure 11, after exposure of SGC-7901 cells to the complexes, the expression of cleaved PARP was observed. In addition, Figure 11 shows that the expression levels of phosphorylated p53 (P-p53), Bcl-xL, and Bcl-2 were dramatically decreased when the SGC-7901 cells were treated with **1**, **2**, or **3,** while the expression level of Bax increased. Under the same conditions, an apparent increase in the expression of caspase-3 was also observed. Thus, our mechanism studies show that **1**–**3** may exert anti-proliferative effects by inducing an apoptotic cascade, which is associated with mitochondrial damage in SGC-7901 cells.

## 4. Conclusions

Three iridium(III) complexes containing –NH_2_ or –NO_2_ groups were synthesized, purified, and characterized. An in vitro cytotoxicity test demonstrated that all three iridium(III) complexes exert moderate anti-proliferative activity against the SGC-7901 cell line, especially the complex **1**, which presented the same cytotoxicity as cisplatin. A subcellular distribution assay showed that **1**–**3** localizes selectively in the mitochondria. Meanwhile, the Ir(III) complexes induce apoptosis in cells through increasing intracellular ROS accumulation, further enhancing intracellular Ca^2+^ levels, and reducing the mitochondrial membrane potential, activating a series of apoptotic responses triggered by caspase family proteins. Moreover, we further evaluated complex-mediated activation of caspase apoptosis, including upregulation of the expression of caspase 3 and downregulation of the Bcl-xl and Bcl-2 proteins. With the intervention of the iridium(III) complexes, the cell cycle was arrested in either the G0/G1 or S phase. It is worth noting that these complexes also inhibited the metastasis of cancer cells, and that the inhibition rate induced by **1** reached 51.09%. In summary, these complexes cause apoptosis through the induction of ROS-mediated mitochondrial dysfunction (Figure 12). In conclusion, this work presents information that is helpful for the design and synthesis of new iridium(III) complexes as potential anti-cancer drug candidates.

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
