# Peer review of "Design, Synthesis, and Anticancer Effect Studies of Iridium(III) Polypyridyl Complexes against SGC-7901 Cells"

_molecules, 2019, doi:10.3390/molecules24173129_

Round 1
Reviewer 1 Report
The manuscript describes the synthesis, characterization and especially the studies of in vitro cytotoxicity and various aspects connected with the mechanism of action for three cyclometalated Ir(III) complexes of the [Ir(ppy)2(L)]+ general formula, where L stands for three different 1,10-phenanthroline derivatives. Although the SGC-7901 cells are given in the title of the manuscript, six cancer cell lines in total were used in the study, complemented by the data from the non-cancerous NIH-3T3 cells. Especially the biological part of the manuscript is sound with a number of interesting results. However, several comments have to be answered before the manuscript will be ready for acceptance:
1/ Above all, the logic of the biological part of the text is, in my opinion, hard to follow. For example, in section 2.3. the cells are treated by the concentrations determined in the following section 2.4! The related processes/properties can be found at different part of the text, such as Inhibition of Colony Formation (2.3.) and Migration Assay (2.12.) or MMP damage (2.8.) and ROS induction (2.10.). This has to be improved in order to make the readability of the revised text as good as possible.
2/ In connection with comment 1 – the authors localized complexes in mitochondria and studied the nuclear DNA damage, which should be explained better. Is DNA damaged directly by complexes (see Fig. 11) or by the enhanced ROS population (see Conclusions)?
3/ The authors studied cytotoxicity and lipophilicity, but their correlation is usually very weak to explain the differences in biological activity. That is why the cell uptake studies should be added to the revised text to see whether this could be a cause of different cytotoxicity of complexes 1-3.
4/ Cytotoxicity was studied at 48 h exposure time (please add to Table 1), but the treatment in the following experiments was sometimes 24 h, sometimes 48 h – please explain why? Especially in the case of cell cycle experiments, the shorter exposure time does not make sense in connection with the typical cell cycle lengths of cancer cells.
5/ The authors claimed (section 3.8.) that “Annexin-V positive cells were considered apoptotic.” However, the discussed results (section 2.5.) reflects only cells in early apoptosis. This has to be corrected and LA populations has to be counted as well.
6/ The complexes exhibit the green fluorescence, which possibly can make a contribution to the obtained flow cytometry signals. Did the authors involve the negative control experiments with a) complexes and b) cells and complexes without the used dyes? This should be added at least to the ROS induction FC experiments.
7/ “Cause and effect” is always a question for similar complexes – it should be discussed (in connection with relevant literature) whether complexes induce ROS which subsequently attack mitochondria, or complexes damage mitochondria, whose dysfunction results to higher ROS population. This also has to be reflected properly in Fig. 11.
8/ Line 174 – should be Ir-2 in my opinion.
9/ Line 233 – what was the level of significance for this statistical evaluation?
10/ Complexes 1 and 3 are known in the literature and can be found in databases, thus only the complex 2 represents a new chemical compound, which has to be highlighted in the revised manuscript for the readers. Although complex 3 was synthesized according the reported procedure, its composition and purity has to be checked by relevant techniques – add at least the results of elemental analysis and 1H NMR.
11/ The discussion of chemistry is poor and I recommend the discussion of the obtained NMR results should be added to the revised text.
12/ The authors should follow the IUPAC recommendations for the nomenclature of the used ligands.
13/ Generally said, there is a bit of controversy about the 2-phenylpyridine labelling in the literature, which, however, can be also found in this manuscript. If electroneutral 2-phenylpyridine is labelled ppy, than the charge of iridium in the [Ir(ppy)2(L)]+ complex cation containing another electroneutral ligand L has to be +I. Two ways how to solve this: a) label 2-phenylpyridine as Hppy, then the used [Ir(ppy)2(L)]+ formula is correct for Ir(III) complexes 1-3; b) if the label for electroneutral 2-phenylpyridine is still ppy, then change the general formula to [Ir(ppy–)2(L)]+ to make it correct for Ir(III) complexes.
14/ Labelling of complexes – why Ir-1-Ir-3 instead of easier and clearer 1-3, especially in the case when only Ir complexes were studied?
15/ Abstract is too wordy, I think that the separate description of the used experiments and their results is not necessary, thus the three sentences “The location of these complexes ... signalling pathway markers.” can be removed from Abstract.
16/ Check the 13C resonance frequency.
17/ English – seems to be with a lot of unacceptable grammar mistakes and typos (e.g. ... may be develop or ... ligand were etc.) and in my opinion it´d be helpful to be checked by a native speaker.
Author Response
Reviewer 1# The manuscript describes the synthesis, characterization and especially the studies of in vitro cytotoxicity and various aspects connected with the mechanism of action for three cyclometalated Ir(III) complexes of the [Ir(ppy)2(L)]+ general formula, where L stands for three different 1,10-phenanthroline derivatives. Although the SGC-7901 cells are given in the title of the manuscript, six cancer cell lines in total were used in the study, complemented by the data from the non-cancerous NIH-3T3 cells. Especially the biological part of the manuscript is sound with a number of interesting results. However, several comments have to be answered before the manuscript will be ready for acceptance:
1/ Above all, the logic of the biological part of the text is, in my opinion, hard to follow. For example, in section 2.3. the cells are treated by the concentrations determined in the following section 2.4! The related processes/properties can be found at different part of the text, such as Inhibition of Colony Formation (2.3.) and Migration Assay (2.12.) or MMP damage (2.8.) and ROS induction (2.10.). This has to be improved in order to make the readability of the revised text as good as possible.
The order has been revised.
2/ In connection with comment 1 – the authors localized complexes in mitochondria and studied the nuclear DNA damage, which should be explained better. Is DNA damaged directly by complexes (see Fig. 11) or by the enhanced ROS population (see Conclusions)?
We have revised the apoptotic mechanism. DNA damage was used to further verify the apoptosis induced by the complexes. DNA damage is directly caused by the complexes.
3/ The authors studied cytotoxicity and lipophilicity, but their correlation is usually very weak to explain the differences in biological activity. That is why the cell uptake studies should be added to the revisedtext to see whether this could be a cause of different cytotoxicity of complexes 1-3.
The cell uptake has been added. However, we can not perform the quantitatively analysis of the cell uptake owing to lack of ICP-MS.
4/ Cytotoxicity was studied at 48 h exposure time (please add to Table 1), but the treatment in the following experiments was sometimes 24 h, sometimes 48 h – please explain why? Especially in the case of cell cycle experiments, the shorter exposure time does not make sense in connection with the typical cell cycle lengths of cancer cells.
The 48 h has been added in Table 1. Most biological experiments were performed for 24 h. However, for cell cycle arrest, we carried out at 24 h and 48 h, there is no obvious effect of the complexes on the cell cycle arrest at 24 h, therefore, we listed the data at 48 h.
5/ The authors claimed (section 3.8.) that “Annexin-V positive cells were considered apoptotic.” However, the discussed results (section 2.5.) reflects only cells in early apoptosis. This has to be corrected and LA populations has to be counted as well.
The percentages of early and late apoptotic cell have been added in the apoptosis section
6/ The complexes exhibit the green fluorescence, which possibly can make a contribution to the obtained flow cytometry signals. Did the authors involve the negative control experiments with a) complexes and b) cells and complexes without the used dyes? This should be added at least to the ROS induction FC experiments.
The complexes exhibit green fluorescence, which has no influence on the flow cytometry signals, because the complexes emit green fluorescence at 600-630 nm, whereas the dye for ROS at 525 nm. The negative control experiments with a) complexes and b) cells and complexes without the used dyes have been added.
7/ “Cause and effect” is always a question for similar complexes – it should be discussed (in connection with relevant literature) whether complexes induce ROS which subsequently attack mitochondria, or complexes damage mitochondria, whose dysfunction results to higher ROS population. This also has to be reflected properly in Fig. 12.
Fig. 12 (mechanism) has been revised, and the relationship has been discussed.
8/ Line 174 – should be Ir-2 in my opinion.
Change Ir-3 into 2
9/ Line 233 – what was the level of significance for this statistical evaluation?
There is no a level of significance for ATP decrease.
10/ Complexes 1 and 3 are known in the literature and can be found in databases, thus only the complex 2 represents a new chemical compound, which has to be highlighted in the revised manuscript for the readers. Although complex 3 was synthesized according the reported procedure, its composition and purity has to be checked by relevant techniques – add at least the results of elemental analysis and 1H NMR.
The data for elemental analysis and ESI-MS have been added. The 1H NMR data have no difference with the values reported by literature (Chao, Biomaterials, 2015, 58, 72-81). Therefore, the data have not been listed.
11/ The discussion of chemistry is poor and I recommend the discussion of the obtained NMR results should be added to the revised text.
The discussion for the NMR has been added to the revised text.
12/ The authors should follow the IUPAC recommendations for the nomenclature of the used ligands.
The ligands have been named according to IUPAC
13/ Generally said, there is a bit of controversy about the 2-phenylpyridine labelling in the literature, which, however, can be also found in this manuscript. If electroneutral 2-phenylpyridine is labelled ppy, than the charge of iridium in the [Ir(ppy)2(L)]+ complex cation containing another electroneutral ligand L has to be +I. Two ways how to solve this: a) label 2-phenylpyridine as Hppy, then the used [Ir(ppy)2(L)]+ formula is correct for Ir(III) complexes 1-3; b) if the label for electroneutral 2-phenylpyridine is still ppy, then change the general formula to [Ir(ppy–)2(L)]+ to make it correct for Ir(III) complexes.
In this manuscript, we used [Ir(ppy)2L]+or [Ir(ppy)2L](PF6)
14/ Labelling of complexes – why Ir-1-Ir-3 instead of easier and clearer 1-3, especially in the case when only Ir complexes were studied?
Change Ir-1-Ir-3 into 1-3.
15/ Abstract is too wordy, I think that the separate description of the used experiments and their results is not necessary, thus the three sentences “The location of these complexes ... signalling pathway markers.” can be removed from Abstract.
The abstract has been revised.
16/ Check the 13C resonance frequency.
In complex [Ir(ppy)2(NAP)](PF6), change C NMR, change 125 MHz into 100 MHz.
17/ English – seems to be with a lot of unacceptable grammar mistakes and typos (e.g. ... may be develop or ... ligand were etc.) and in my opinion it´d be helpful to be checked by a native speaker.
English has been revised.
Reviewer 2 Report
This study describes the synthesis of three iridium(Ⅲ) complexes and the investigation of their cytotoxicity against a series of cancer and normal NIH 3T3 cells showing moderate in vitro antitumor activity toward SGC-7901 cells. The complexes effectively inhibit the metastasis and proliferation of tumor cells inducing apoptosis in SGC-7901 cells through ROS-mediated mitochondrial damage and DNA damage pathways.
Overall, the work appears to be of good quality and the manuscript is well presented. In my opinion, the paper is of interest for readers of Molecules with minor revision.
In particular, the authors should discuss the biological data in comparison with the free ligands and the starting material [Ir(ppy)2Cl]2.
Author Response
Reviewer 2# This study describes the synthesis of three iridium(Ⅲ) complexes and the investigation of their cytotoxicity against a series of cancer and normal NIH 3T3 cells showing moderate in vitro antitumor activity toward SGC-7901 cells. The complexes effectively inhibit the metastasis and proliferation of tumor cells inducing apoptosis in SGC-7901 cells through ROS-mediated mitochondrial damage and DNA damage pathways.
Overall, the work appears to be of good quality and the manuscript is well presented. In my opinion, the paper is of interest for readers of Molecules with minor revision.
In particular, the authors should discuss the biological data in comparison with the free ligands and the starting material [Ir(ppy)2Cl]2.
The cytotoxicity of the free ligand and starting material [Ir(ppy)2Cl]2has been added and discussed in the cytotoxic activity section.
Reviewer 3 Report
This paper is about the synthesis, characterization and biological activity of three polypyridil Ir(III) complexes, although one of them (Ir-3), at least from what one can infer by the experimental part, had been previously prepared and characterized by a different group and already reported in the literature.
The other two complexes have been briefly described, their NMR and MS data are reported in the experimental part, while the UV-vis results have not been discussed. Only the relative spectra are shown, together with luminescence spectra, poorly discussed as well. To my opinion, the description of the complexes can be improved.
The work has been designed to be complete and exhaustive, still the reading of the manuscript is slightly unpleasant. The experimental part has been located at the end of the paper, so that one has to move back and forth to recover details useful for the comprehension of the work. Moreover, some experimental details have been repeated in the “result and discussion” section, in order to make it more understandable, I guess. Also some acronyms become clear only after reading the experimental part, that is at the end of the paper.
One more thing is the choice of different concentration for the three complexes which have been used for the biological tests. Ir-1 was 4 μM, while the other two were 15 μM. I had to use my fantasy to suppose that maybe this could be related to the IC50 values of the three species, as reported in table 1 (3.6, 14.1 and 11.1 μM, respectively). If this is the case, I find it rather unusual, since it is normal to compare the effect of species in the same conditions (also concentration), and however it should be specified before starting the discussion. As I find unusual that the results of the colony formation assay were taken after 4 days.
Overall, the manuscript could have been written in a shorter and better way, taking into account that also the English grammar has to be improved in some parts.
Author Response
Reviewer 3# This paper is about the synthesis, characterization and biological activity of three polypyridil Ir(III) complexes, although one of them (Ir-3), at least from what one can infer by the experimental part, had been previously prepared and characterized by a different group and already reported in the literature.
The other two complexes have been briefly described, their NMR and MS data are reported in the experimental part, while the UV-vis results have not been discussed. Only the relative spectra are shown, together with luminescence spectra, poorly discussed as well. To my opinion, the description of the complexes can be improved.
The UV-Vis and luminescence spectra of the complexes have been discussed.
The work has been designed to be complete and exhaustive, still the reading of the manuscript is slightly unpleasant. The experimental part has been located at the end of the paper, so that one has to move back and forth to recover details useful for the comprehension of the work. Moreover, some experimental details have been repeated in the “result and discussion” section, in order to make it more understandable, I guess. Also some acronyms become clear only after reading the experimental part, that is at the end of the paper.
To easy reading for reader, we place materials and methods before the results and discussion, we have also revised the order.
One more thing is the choice of different concentration for the three complexes which have been used for the biological tests. Ir-1 was 4 μM, while the other two were 15 μM. I had to use my fantasy to suppose that maybe this could be related to the IC50 values of the three species, as reported in table 1 (3.6, 14.1 and 11.1 μM, respectively). If this is the case, I find it rather unusual, since it is normal to compare the effect of species in the same conditions (also concentration), and however it should be specified before starting the discussion. As I find unusual that the results of the colony formation assay were taken after 4 days.
If the concentrations of three complexes are the same, complexes 2 and 3 will show low ROS, mitochondrial membrane potential, etc. Hence, we used the same concentration to perform the biological experiments. We carried out the colony after SGC-7901 cells were exposed to the complexes for 8 days.
Overall, the manuscript could have been written in a shorter and better way, taking into account that also the English grammar has to be improved in some parts.
English grammar has been improved in the whole manuscript.
Round 2
Reviewer 1 Report
The manuscript was improved and the authors showed much effort to address the comments of the reviewer (e.g. reorganization of the text or cell uptake experiments). Nevertheless, I´m not fully satisfied with two issues, which have to be corrected before the revised text will be ready for the acceptance:
13/ Generally said, there is a bit of controversy about the 2-phenylpyridine labelling in the literature, which, however, can be also found in this manuscript. If electroneutral 2-phenylpyridine is labelled ppy, than the charge of iridium in the [Ir(ppy)2(L)]+ complex cation containing another electroneutral ligand L has to be +I. Two ways how to solve this: a) label 2-phenylpyridine as Hppy, then the used [Ir(ppy)2(L)]+ formula is correct for Ir(III) complexes 1-3; b) if the label for electroneutral 2-phenylpyridine is still ppy, then change the general formula to [Ir(ppy–)2(L)]+ to make it correct for Ir(III) complexes.
Re: In this manuscript, we used [Ir(ppy)2L]+or [Ir(ppy)2L](PF6)
Comment: I´ll try to explain it differently. If you write ([Ir(ppy)2(L)](PF6) and define ppy = 2-phenylpyridine and L = phenanthroline derivative, than the oxidation state of Ir has to be +I, because you have three electroneutral ligands in the complex cation and monoanion PF6. On the other hand, if 2-phenylpyridine is labelled as Hppy, then ppy means deprotonated ligand with 1- charge, and in this case the formula [Ir(ppy)2(L)](PF6) fits for the Ir(III) complex. I recommend to leave the formulas unchanged and define 2-phenylpyridine as Hppy in the revised manuscript.
16/ Check the 13C resonance frequency.
Re: In complex [Ir(ppy)2(NAP)](PF6), change C NMR, change 125 MHz into 100 MHz.
Comment: The resonance frequency of 13C atoms is some 125 MHz when measured by the 500 MHz device (not 100 MHz, 125 Hz or 100 Hz).
Author Response
The manuscript was improved and the authors showed much effort to address the comments of the reviewer (e.g. reorganization of the text or cell uptake experiments). Nevertheless, I´m not fully satisfied with two issues, which have to be corrected before the revised text will be ready for the acceptance:
13/ Generally said, there is a bit of controversy about the 2-phenylpyridine labelling in the literature, which, however, can be also found in this manuscript. If electroneutral 2-phenylpyridine is labelled ppy, than the charge of iridium in the [Ir(ppy)2(L)]+ complex cation containing another electroneutral ligand L has to be +I. Two ways how to solve this: a) label 2-phenylpyridine as Hppy, then the used [Ir(ppy)2(L)]+ formula is correct for Ir(III) complexes 1-3; b) if the label for electroneutral 2-phenylpyridine is still ppy, then change the general formula to [Ir(ppy–)2(L)]+ to make it correct for Ir(III) complexes.
Re: In this manuscript, we used [Ir(ppy)2L]+or [Ir(ppy)2L](PF6)
Comment: I´ll try to explain it differently. If you write ([Ir(ppy)2(L)](PF6) and define ppy = 2-phenylpyridine and L = phenanthroline derivative, than the oxidation state of Ir has to be +I, because you have three electroneutral ligands in the complex cation and monoanion PF6. On the other hand, if 2-phenylpyridine is labelled as Hppy, then ppy means deprotonated ligand with 1- charge, and in this case the formula [Ir(ppy)2(L)](PF6) fits for the Ir(III) complex. I recommend to leave the formulas unchanged and define 2-phenylpyridine as Hppy in the revised manuscript.
Change ppy into Hppy
16/ Check the 13C resonance frequency.
Re: In complex [Ir(ppy)2(NAP)](PF6), change C NMR, change 125 MHz into 100 MHz.
Comment: The resonance frequency of 13C atoms is some 125 MHz when measured by the 500 MHz device (not 100 MHz, 125 Hz or 100 Hz).
Change 100 Hz into 125 MHz
Reviewer 3 Report
The authors tried to do their best to correct the paper as suggested by the reviewers. Nevertheless, they did not solve a couple of the observations.
For instance, a single line was added in the attempt to discuss NMR characterization of the complexes: “In the spectra of 9.46 (s, 1H) for 1 is attributed to the proton of Ca, the peak of 6.24 (t, 2H) for 2 is assigned to the protons of -NH2 group”. Apart from being not clear in its meaning, this sentence does not represent a proper discussion on the changes undergone by the signals of ligands upon complexation. For this reason, a comparison between the resonances of the free ligands and the complexes should be at least added under the form of a table, if the authors do not want to report a full description in section 3.1, “synthesis and characterization”.
The fact that the concentrations of the different complexes in the biological tests was not the same (Ir-1 was 4 μM, while the other two were 15 μM) has not been justified in the text, I think a line to explain this could be useful for the reader, as it was to me when I found it in the authors’ reply.
There are still a number of grammar and typing mistakes.
I would recommend to solve these problems before publication of the manuscript.
Author Response
The authors tried to do their best to correct the paper as suggested by the reviewers. Nevertheless, they did not solve a couple of the observations.
For instance, a single line was added in the attempt to discuss NMR characterization of the complexes: “In the spectra of 9.46 (s, 1H) for 1 is attributed to the proton of Ca, the peak of 6.24 (t, 2H) for 2 is assigned to the protons of -NH2 group”. Apart from being not clear in its meaning, this sentence does not represent a proper discussion on the changes undergone by the signals of ligands upon complexation. For this reason, a comparison between the resonances of the free ligands and the complexes should be at least added under the form of a table, if the authors do not want to report a full description in section 3.1, “synthesis and characterization”.
A comparison between the resonance of ligands and the complexes have been added in the manuscript.
The fact that the concentrations of the different complexes in the biological tests was not the same (Ir-1 was 4 μM, while the other two were 15 μM) has not been justified in the text, I think a line to explain this could be useful for the reader, as it was to me when I found it in the authors’ reply.
In the cytotoxicity in vitro, we add the sentence “In the biological activity assays, we used near IC50 values as concentration of the complexes, namely, 4.0 µM for 1, 15.0 µM for 2 and 3, respectively.”
There are still a number of grammar and typing mistakes.
We have carefully checked the manuscript, and some grammar and typing mistakes have been revised.
I would recommend to solve these problems before publication of the manuscript.